# Impact of Common Mistletoe (*Viscum album* L.) on Scots Pine Forests—A Call for Action

**Hanna Szmidla** [1],* [iD], **Miłosz Tkaczyk** [1], **Radosław Plewa** [1] [iD], **Grzegorz Tarwacki** [1] [iD] and **Zbigniew Sierota** [2]

[1] Department of Forest Protection, Forest Research Institute, Braci Leśnej 3, Sękocin Stary, 05-090 Raszyn, Poland; M.Tkaczyk@ibles.waw.pl (M.T.); R.Plewa@ibles.waw.pl (R.P.); G.Tarwacki@ibles.waw.pl (G.T.)

[2] Department of Forestry and Forest Ecology, Faculty of Environmental Management and Agriculture, University of Warmia and Mazury in Olsztyn, Pl. Łódzki 2, 10-727 Olsztyn, Poland; zbigniew.sierota@uwm.edu.pl

\* Correspondence: h.szmidla@ibles.waw.pl; Tel.: +48-227-150-353

**Abstract:** Common mistletoe is increasingly mentioned as contributing not only to the decline of deciduous trees at roadside and in city parks, but to conifers in stands. The presence of *Viscum* in fir stands has been known for many years, but since 2015 has also been the cause of damage to pine. In 2019, mistletoe was observed on 77.5 thousand hectares of Scots pine stands in southern and central Poland. Drought resulting from global climate change is implicated as an important factor conducive to weakening trees and making them more susceptible to the spread of mistletoe and other pests. This paper presents an overview of the latest information on the development of this semi-parasitic plant in Poland, its impact on tree breeding traits and raw material losses, as well as current options for its prevention and eradication.

**Keywords:** common mistletoe; water stress; pine tree dieback; timber losses

## 1. Is Mistletoe a Problem in The Forests?

The effects of *Viscum* spp. infection include reduced tree vitality, shoot die-off, and reduction of the quality and volume of wood produced (Figure 1). The influence of common mistletoe (*Viscum album* L.) on deciduous trees, especially along roadsides, in parks, and in plantations, has been well studied for years [1–3]. According to Barney et al. [4], mistletoe colonizes over 450 species and varieties of trees. Mistletoe can reduce flowering and fruiting and increase susceptibility to damage from insects and fungi, which can result in premature death of commercially valuable trees [5,6]. Similar consequences for tree development are caused by the semi-parasitic African mistletoe (*Loranthus micranthus* Linn.), which depends on water and nutrients from the host plant, although the mistletoe produces its own carbohydrates by photosynthesis [7]. Semi-parasitic species of mistletoe are also found in North America (fir dwarf mistletoe, *Arceuthobium abietinum* Engelm. ex Munz) [8], in Asia (Korean mistletoe, *Viscum coloratum* (Komarov) Nakai) [9], and in South America the aerial parasite, *Tristerix corymbosus* (L.) Kuijt (Loranthaceae) [10]. The genus *Viscum* includes approximately 100 species, most occurring in Africa and Madagascar, with a smaller number in southern Asia and only a few species in Europe, including *Viscum album* L. [11].

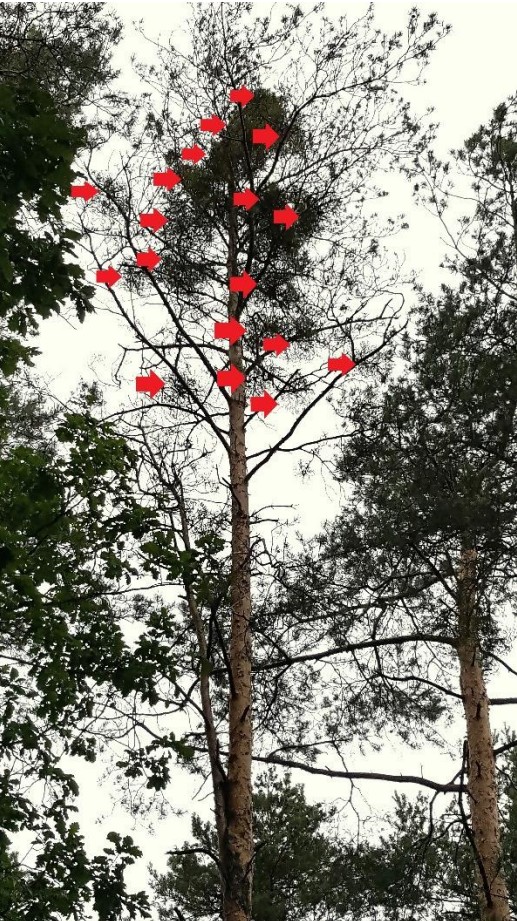

**Figure 1.** Scots pine tree with 18 mistletoes (*V. album* subsp. *austriacum*) in the defoliated crown.

Within the *Viscum* species a number of subspecies have been distinguished. Due to the high morphological similarity, the easiest way is to distinguish them is on the basis of the host plants they parasitize. On this basis, four *Viscum* subspecies are found in Europe:

- *V. album* subsp. *album*, which parasites deciduous trees and shrubs [11];
- *V. album* subsp. *austriacum*, which is found only on the genera *Pinus* and *Picea* [11];
- *V. album* subsp. *abietis*, which occurs exclusively on fir (*Abies* spp.) [12]; and
- *V. album* subsp. *creticum*, which develops only on the Calabrian pine (*Pinus brutia*) in Crete [13].

Among these subspecies, two have gained prominence: *V. album* subsp. *austriacum* and *V. album* subsp. *abietis*. According to ICP Forests [14], in European Scots pine (*Pinus sylvestris* L.) stands, pine mistletoe (*V. album* subsp. *austriacum*) was the most frequent cause of biotic damage (7.4%), followed by needle cast/needle rust fungi (6.5%).

*Viscum album* subsp. *abietis* (white mistletoe) is one of the most significant biotic factors affecting Silver fir in natural forests. In the last 60 years in Europe, many reports were made about mistletoe affecting Silver fir: in France [15,16]; in Switzerland in the 1970's, 1980's, and 2000's [17,18]; in Croatia [19]; in the Spanish Pyrenees [20]; and in the Romanian Carpathians [21]. Mistletoe is also a major problem in the fir forests on Mount Parnis, Greece [22]. More than 30% of firs in some regions of Croatia (e.g., in Gorski Kotar, 32.8%) were infected by mistletoe [19], and in the Romanian Carpathians the infection rate reaches 42% [21]. There is a lack of consensus regarding the effects of *V. album* subsp. *abietis* on the health of fir. In some studies, white mistletoe was considered a parasite causing large losses in wood production [23,24] and killing entire stands of *Abies* [15,25]. Other studies suggest that heavy infestations result in mortality of a limited number of individuals [17,22,26]. In addition, the

incidence of infection may be spreading to new areas, since the elevation at which mistletoe occurs has shifted 200 m uphill in the Alps during the last century. It likely happens due to an increase of temperature in winter season [27], and the minimum temperature in winter is one of the main limiting factors for mistletoe [28]. Also, the fact that with climate change there will be more susceptible hosts at warmer sites can explain the shift in altitude [27].

In Austrian pine stands (*Pinus nigra* J.F. Arnold), mistletoe accounted for 12.6% of damage caused by biotic factors [14]. In recent years in some jurisdictions, *V. album* subsp. *austriacum* has spread rapidly, such as in the German federal state of Brandenburg, where mistletoe infection rate of pine trees increased from 1% to 11% between 2009 and 2015 [29]. In contrast, in the Czech Republic, mistletoe on conifers is not considered significant, so that attention is paid only to infection of deciduous trees. In Slovakia, mistletoe has been observed in pine and fir stands, but the damage is not monitored (personal information).

In Poland, the presence of mistletoe on roadside and park trees is common, but the spread of *Viscum album subsp. austriacum* in forests is of concern. According to the report "Short-term forecast of the occurrence of major pests and infectious diseases of forest trees in 2019" [30], in 2017 1.4 thousand hectares of coniferous forest were affected by mistletoe, but in 2018, due to drought and greater transparency of the tree crowns (making mistletoe in trees easier to see), the area noted increased to almost 23.0 thousand ha. Its occurrence was concentrated in the southern and central parts of Poland. In 2019, a new mistletoe identification system was implemented in Polish State Forests Holding, covering an area of 9.2 million hectares, 60.1% of which is Scots pine. The severity of infection was shown by the fact that at least 30% of surveyed trees had at least one mistletoe, resulting in an estimated area of 77.5 thousand hectares of Scots pine stands affected across more than 330 Forest Districts with *Viscum*.

A climatic basis for the spread of *Viscum* in Poland is suggested by the geographic distribution of pine mistletoe in Scots pine stands in 2019 (Figure 2c), compared to changes in temperature and precipitation (percent change in long-term means of air temperature and precipitation in spring 2018) (Figure 2a,b), as follows:

- the largest area of affected stands occurs in the south of the country, with affected areas running in a band towards the north-east, in the direction of prevailing warm winds;
- the distribution of mistletoe in Scots pine stands occurs mostly in areas with the highest percentage increase in spring air temperature and decrease in precipitation; presumably, these conditions increase stress in trees which favors *Viscum* colonization and creates tree crown conditions conducive for birds to act as vectors of seeds.

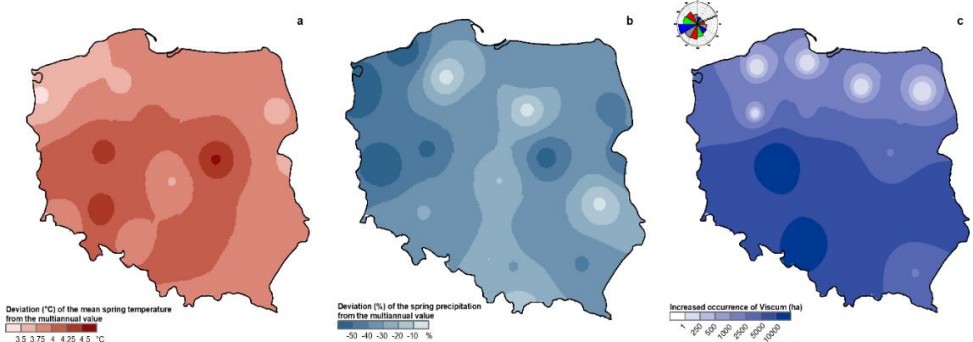

**Figure 2.** The spatial distribution of percentage deviation of spring air temperature from the multiannual mean (1971–2000) (**a**), percentage deviation of spring precipitation (**b**), and area of Scots pine stands in Poland with visible *V. album* subsp. *austriacum* in 2019 (**c**), with dominant wind directions (c, left corner); white space in (**c**) means no mistletoe presence or no data.

A correlation test was conducted to compare the area of Scots pine stands affected by mistletoe and the value of the average Sielianinov's hydrothermal coefficient (HTC) [30]. The hydrothermal coefficient was calculated as:

$$HTC = P \times 10 / \left( \sum t \right)$$

where P is the sum of monthly precipitation and Σt is the sum of average monthly temperatures during the vegetation period (from April to October) 2010–2018. A significant linear relationship was found (Figure 3), such that areas where HTC was lower had a larger area affected by mistletoe.

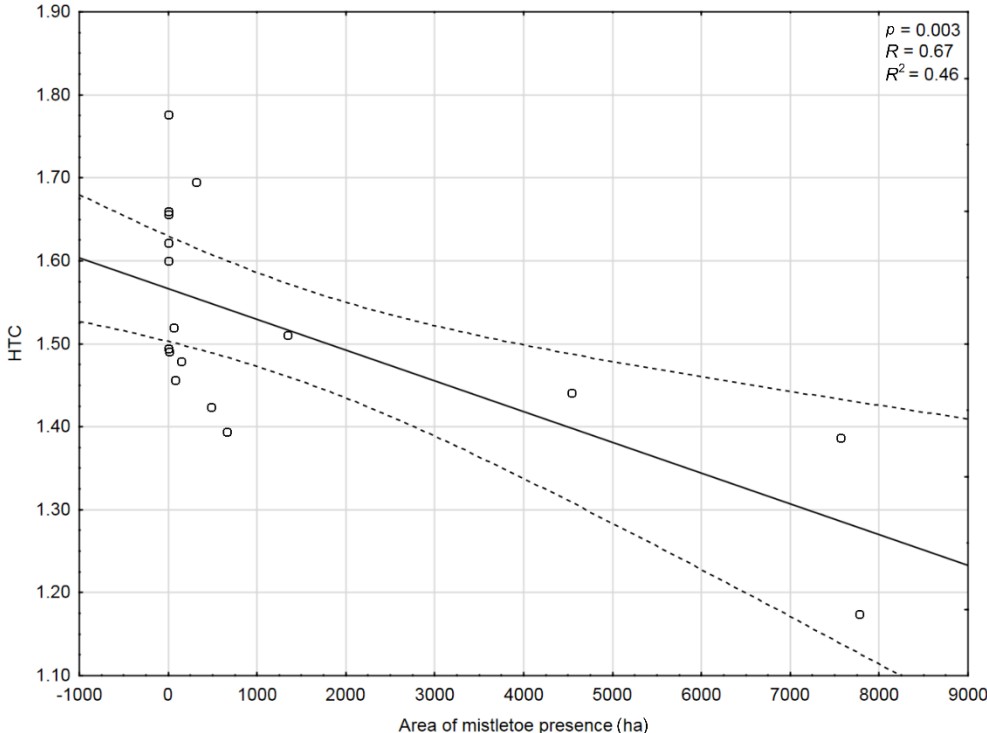

**Figure 3.** Relationship between the hydrothermal coefficient (HTC) during the vegetation period 2010–2018 and the occurrence of pine mistletoe (*V. album* subsp. *austriacum)* in Scots pine stands in 2018 (the multiyear average HTC = 1.5).

## 2. *Viscum* Biology and Host Impact

Mistletoe can initiate host infection in several ways. One way comes from "long distance" spread of mistletoe seeds by birds, with the most important bird species being *Turdus viscivorus* L., *T. pilaris* L., *Bombycilla garrulus* L., and *Sylvia atricapilla* L. [11,31,32]. The first three species mentioned ingest the mistletoe berries whole and the seeds then lack their characteristic white skin, which is removed in the digestive tract of the birds. *Sylvia atricapilla*, however, feeds only on the skin and leaves the seed on the branch, near the parent mistletoe bush, where it may germinate [33]. Germination does not require that the seed pass through the bird's digestive system. Wangerin [34] showed that the low nutritional value of mistletoe berries necessitates that birds eat large quantities of seed to meet their calorific needs (e.g., up to 100 berries per day) [11]. This is a key strategy in the survival and spread of *Viscum* species. The initial spread of *Viscum* may be unnoticeable because new germinant growth is very slow, however growth rates increase as mistletoe plants become older [35]. Because birds land at the tops of trees and infections start there, the oldest mistletoe plants are found closer to the top of the host tree. In the case of *P. nigra,* mistletoe can achieve maximum host plant colonization within about 10–15 years of initial infection [36,37].

An evolutionary mechanism facilitating the "short distance" spread of mistletoe is the viscous substance they contain, called viscin. Viscin enables mistletoe seeds to adhere to the host branch,

causing a new infection. Mistletoe fruit that are spontaneously shed from *Viscum* and come in contact with a branch can develop into a new plant. Mistletoe fruit can also stick to the limbs of birds and then be transferred by them over long distances [11].

When mistletoe colonize host tissue, they can produce two types of endophytic systems. The first type is called haustoria, which overgrow the tissues of the host plant, eventually penetrating the host plant's cambium, which allows the mistletoe to absorb water and minerals (this is known as deep penetration). The second type of endophytic system is the production of cortical bands by the mistletoe that penetrate the tree through parenchymal tissues and phloem, where they spread laterally or longitudinally [38,39]. The cortical bands contain chlorophyll and have an average length of 4 to 6 cm [34,40]. Although phloem and xylem are present in these bands, it has yet to be confirmed whether these bands form a physical connection allowing water and nutrients to be absorbed from the host plant [41].

Sexual reproduction of mistletoe usually starts when plants are 4–5 years old. *Viscum album* is dioecious and the distinction between female and male plants is impossible until the plant has blossomed, with the sex system usually deviating from 1:1 [42,43]. Pollination occurs most often by insect vectors [34,44,45]. Although pollination by wind occurs less commonly, pollen can be carried up to 2 km by air [45]. Reproduction also can occur vegetatively through the production of adventitious shoots near the mother bush [45], especially when the plant is damaged mechanically (e.g., by breaking, pruning or freezing). When mistletoe seed is in a resting state, which lasts on average 5 to 6 months in the winter months [34], there is no cell division or DNA synthesis in seed tissues. Seed activity (i.e., cell division and DNA synthesis in the terminal meristem) starts only 3–4 days after germinating [46]. According to Stopp [24], *V. album* can germinate on almost any type of surface, including glass, stone, wood, paper, etc., because only light and temperatures of 8–10 °C are required for germination [47]. The optimum temperature for development is 15–20 °C [48]. Hypocotyl growth can last up to 60 days and during this time it reaches a length of about 5–6 mm [33,41]. The hypocotyl bends towards the host plant and swells to form suckers, which complete the short non-parasitic part of the mistletoe life cycle. The seed dies if the hypocotyl ends up on a dead plant or the wrong species of tree. With a suitable host, the mistletoe cells start to penetrate into the host tissues with the help of enzymes and develop the first haustoria, from which the whole mistletoe plant is formed [49,50].

According to Fisher [51], *Viscum* mistletoe can cause water stress in the host plant, especially in host plant branches acropetal to the infection site. However, one must ask whether the increased presence of mistletoe in coniferous forests in recent years is the result of water stress in host trees due to drought. Furthermore, it can be asked whether an increase in occurrence of drought is at least in part attributable to climate change [27]. Mistletoe parasitically obtains both water and mineral compounds absorbed by the host plant's mycorrhizal roots, while mistletoe generates its own supply of carbohydrates by photosynthesis (Figure 4).

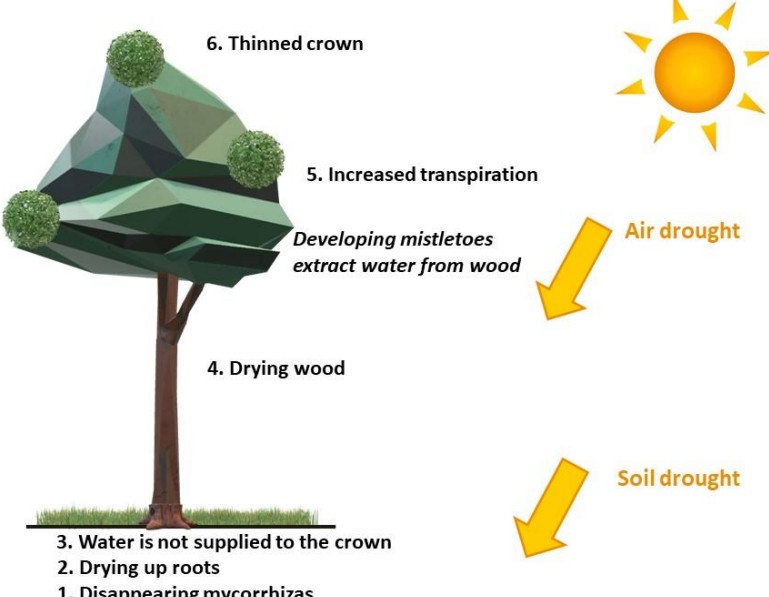

**Figure 4.** Probable course of Scots pine disease involving mistletoe.

The loss of water and mineral nutrients is manifested in host plants by water deficiency, decreased biomass increment, shortened needles, and poorer quality of seeds [52,53]. The rate of transpiration of water in mistletoe is higher than in the host plant [54], which reduces the efficiency of water use by the host by up to 9 times [37,55,56]. Stress resulting from water use by mistletoe causes water deficit in the host, which reduces food resources in needles due to reduced host plant photosynthesis, especially during summer drought [55,57–59]. Furthermore, the increasingly closed stomata (due to water stress) reduce carbon assimilation for the tree [60]. Such conditions favor co-infection, i.e., infection of the tissue of the tree by disease-causing agents that further weaken the host tree. The poorer health of parasitized trees affects their growth, reproduction, and wood quality [52,53].

Nutrients absorbed by trees—macroelements N, K, Ca, P, and S and trace elements Mg, Fe, Cu, Zn, Mo, B, Na, and N—accumulate in mistletoe, making them inaccessible to the host, especially in needles, increasing the adverse impact of drought [57]. It was found that the total chlorophyll content in pine needles affected by mistletoe increased from April to June, then decreased to September, whereas in healthy plants it increased from April to October [56]. In needles of infected pines, the decrease in chlorophyll content resulted in a decrease in photosynthetic efficiency and damage to chloroplasts resulted from $Fe^+$ deficiency [57]. This caused fewer and smaller needles to be produced by mistletoe-parasitized pines compared to non-affected trees.

Mistletoe can affect mycorrhizal associations of the host plant. Sanders et al. [61] found that the parasitic dodder (*Cuscuta* sp.) developed more easily when tree roots are mycorrhizal, because mycorrhizae directly supply water to the plant through the roots. However, the mycorrhizal effect on dodder growth occurred before the haustoria of dodder had succeeded in penetrating the host. These results indicate that colonization by mycorrhizal fungi had systemic effects on their hosts, which altered either the nature of prepenetration dodder signals or the levels of nutrients contained in host stem exudates. The presence of *Arceuthobium* in *Pinus contorta* alters ectomycorrhizal fungal community structure and decreases the richness of ectomycorrhiza fungi species, however it does not influence the level of mycorrhization. Ectomycorrhizae species that were less carbon-consuming were dominant, and therefore they were less competitive with root pathogens [62]. On the other hand, Gehring and Whitham [63] found a negative correlation between *Phoradendron juniperinum* and the mycorrhizal community in *Juniperus monosperma* and stated that the presence of this mistletoe reduced mycorrhizal richness by 27%–38%.

According to Gill and Hawksworth [64], *Viscum* makes host plants more susceptible to pathogenic infection, which can increase mortality. However, when Gea-Izquierdo et al. [63] examined mistletoe-infected *Pinus pinaster* Ait. for pathogens, they found no evidence of infection by *Heterobasidion* or *Phytophthora*. The pathogen *Armillaria mellea*, which weakens or causes stress in trees, was isolated in only a few soil samples. Studies by Gea-Izquierdo et al. also showed that the incidence of biotic factors (including pathogens) analyzed in soil, needles, and wood did not differ in dying and dead trees infected by *Viscum,* compared with healthy trees. There was also no difference in the occurrence of secondary pests. Only about 50% of trees on which strong defoliation was found were infected by mistletoe [65]. Defoliation was found to be dictated more by drought, leading to the slow death of pine trees, with the presence of mistletoe being secondary [65,66].

## 3. Potential Biocontrol Agents

Biocontrol is now a core component of integrated pest management. Considerable success has been achieved in the implementation of biological control strategies in agriculture, forestry, and greenhouse horticulture [67]. Biocontrol is defined as "the study and uses of parasites, predators, and pathogens for the regulation of host (pest) densities" [68]. This method has gained acceptance for control of parasitic plants using pathogens due to its practicality, safety, and environmental benefits.

Several fungi which infect mistletoes are potential biological control agents (BCAs): *Plectophomella visci* (Sacc.) Moesz, *Septoria visci* Bres. and *Sphaeropsis visci* Alb. & Schwein.) Sacc. [69], *Colletotrichum gloeosporoides* (Sacc.) Penz. [70], *Botryosphaerostroma visci* (Plectophomella visci Moesz) [50,71,72], *Botryosphaeria dothidea* (Moug. Fr.) Ces. & De Not., *Gibberidea visci* (Fuckel) [73] and *Botryosphaeria visci* (Kalchbr.) Arx & E. Mull [74], *Alternaria alternata* (Fr.) Keissl. and *Acremonium kiliense* Grütz [75].

Parasitic bacteria can also be used as BCA are. Kotan et al. [75] investigated five bacterial strains, including two types of *Burkholderia cepacia* and one of each of *Bacillus megaterium, Bacillus pumilus*, and *Pandoraea pulmonicola*. These bacterial strains were pathogenic to *Viscum* when applied by injection, but none were pathogenic when sprayed on mistletoe.

*V. album* is known to be affected by relatively few pathogens, presumably because it possesses an effective defence system [76]. Although there are many reports of the isolation of pathogens found to be attacking mistletoes, none have been developed for operational use as BCAs [72].

In Europe, several taxa of insects are reportedly found on mistletoe shoots. The most common species of beetles developing in mistletoe belong to the jewel beetle family (Buprestidae): *Agrilus viscivorus* Bílý, 1991, *A. graecus* Obenberger, 1916, *A. jacetanus* Sánchez & Tolosa, 2004 [77]. All these species feed on mistletoe, and are capable of killing it and limiting its population. Mistletoe are also targets of *Agrilus kutahyanus* Królik, 2002, a non-native beetle from Asia (Turkey) [78]. All these beetle species colonize mistletoe shoots, feeding on them and potentially leading to mistletoe mortality. Other species of Coleoptera in Poland and other European countries are rarely found on mistletoe, for example: *Gastrallus knizeki* (Zahradník, 1996) (Ptinidae), *Ptinomorphus imperialis* (Linnaeus, 1767) (Ptinidae), *Rhaphitropis marchica* (Herbst, 1797) (Anthribidae) *Lathropus sepicola* (P.W.J. Müller, 1821) (Laemophloeidae), *Arthrolips nana* (Mulsant et Rey, 1861) (Cerylophidae) [79–81], *Pogonocherus hispidus* (Linnaeus, 1758) (Cerambycidae) [82], *Oplosia cinerea* (Mulsant, 1839) (Cerambycidae) [83], *Xylosandrus germanus* (Blandford, 1894) (Curculionidae: Scolytinae) [84], and *Ixapion variegatum* (Wencker, 1864) (Apionidae) [85].

Only a few species of moths develop on mistletoe. For example, *Synanthedon loranthi* (Králíček, 1966) develop in mistletoe that has parasitized Scots pine. The larvae overwinter inside the mistletoe twigs and pupate on the surface of the mistletoe [86]. Understanding the biology and behavior of this moth species has contributed to many discoveries of its presence in Europe. Currently, this species is found in the central and southern parts of the continent [87,88].

Two other species of moth have been discovered that feed on the leaf blades of mistletoe. These species belong to the tortrix moth family (Tortricidae): *Celypha woodiana* (Barrett, 1882) [89] and polyfag red-barred Tortrix: *Ditula angustiorana* (Haworth, 1811) [90,91]. Several species of

Hemiptera develop on mistletoes: *Cacopsylla visci* (Curtis, 1835) (Psyllidae), *Anthocoris visci* Douglas, 1889 (Anthocoridae), and *Pinalitus viscicola* (Putton, 1888) (Miridae) [92,93].

## 4. Tree Growth and Economic Losses Caused by Mistletoe and Possibilities of Prevention and Control

Mistletoe infection causes crown thinning by reducing foliage biomass (Figure 3). Bilgili et al. [56] reported that needle biomass may be reduced by as much as 40% in infected trees. Decreased chlorophyll content in foliage of infected trees [58,94] leads to significant reduction of photosynthetic capacity [95] and results in growth losses of host trees [96,97]. Some authors from Germany, Switzerland and Spain reported that *Viscum* spp. reduce radial growth of *P. nigra* and *P. sylvestris*, especially during drought [96,98,99]. Mistletoe reduces the amount of carbon absorbed and decreases host tree carbohydrates by 22%–43%, thus limiting the growth of the host and affecting the quality and quantity of wood produced. Studies on the effect of mistletoe on host tree radial growth [52] showed a significant reduction in the average width of the annual growth ring, compared to non-colonized trees. A significant correlation was also found between growth loss and the degree of pine infection by mistletoe [56]. Parasitized trees have lower levels of photosynthetic activity and overall vitality, as well as reduced seed production [11,53,58,60,100–102].

Quantitative data on the scale of economic losses due to mistletoe are lacking [6,64,103], except for dwarf mistletoes (*Arceuthobium* spp.), which are common and widespread pathogens of commercially valuable conifers [5,104]. Drummond [105] estimates a total annual growth loss (difference between realized volume in an infested stand and potential yield for the site) from dwarf mistletoe in the United States at 11.5 million cubic meters per year. Vázquez [106] reports a loss of 2 million cubic meters per year in Mexico. Estimates for Canada are available for Newfoundland at 1 cubic meter per ha per year [107] and for Manitoba, Saskatchewan, and Alberta at 2.4 cubic meters per hectare per year [108]. Similarly to dwarf mistletoe, the common mistletoe causes economic losses resulting from decreased rate of host tree growth. In addition, it reduces tree health and causes poorer seed production, as well as having indirect ecological impacts, such as its effects on natural regeneration (e.g., an increase in understory species diversity and growth promoted by more light due to canopy thinning and increased soil fertility from organic matter deposition under affected trees) [52,109].

The occurrence of *Viscum album* is related to many factors, such as the vitality and extent of forests, changes in the proportions of forest tree species, changes in many abiotic factors, e.g., changes in general climate, air pollution, droughts [21,110], and human impacts on the environment, (e.g., forest management, cultivation techniques, or planting of alien woody species [2,11,34]). Predicting how climate change will alter tree–parasitic plant relationships is difficult, since climate change can be expected to affect both the host and the parasite species individually, leading to changes in the strength or even presence of the symbiosis. Since rising temperatures and changing precipitation patterns are expected to increase drought frequency and severity in many regions, and plant parasites increase host tree water stress and drought-associated mortality [111,112], infection is likely to exacerbate climate-related drought stress in forests.

Direct mistletoe control is at present the only practical method of reducing infection rates [6]. Direct methods consist of pruning infected branches or removing infected trees. Due to the high labor intensity and costs, these methods are applicable only in small, high value areas, such as parks, orchards, plantations, or single trees in cities [45,49]. The removal of infected branches from Scots pine trees removes a source of internal competition for water and nutrients, leading to increased diameter and height growth of pruned trees [99].

Chemical control of mistletoe is currently used in several non-European countries, e.g., India, Bangladesh [113,114], and Australia [115], but it provides partial success only. Few herbicides are able to selectively control parasitic plants without damaging the host species [116,117]. The systemic herbicides 2.4-D, 2.4-5 T, 2.4-MCPB, and di-chloroethane were found to kill *V. album* subsp. *abietis* on *Abies* with only slight host damage. Tests with these herbicides on *V. album* subsp. *album* growing on

deciduous trees also look promising [6]. Baillon et al. [118] reported detailed experiments with 2.4-DB and glyphosate. They observed that control lasts for 4–6 months after treatment, with no herbicide found in the host.

The application of the plant growth regulator ethephon (2-Chloroethylphosphonic acid) has been examined for mistletoe control by Adams et al. [119]. Ethephon releases ethylene during absorption by the plant, which enhances the natural maturation process and leads to abscission of mature mistletoe shoots. However, the endophytic system of the parasite is not affected and thus ethephon only leads to defoliation, not eradication of the mistletoe. In the Czech Republic, the product Cerone 480 SL, which has ethephon as an active ingredient, has recently been registered for controlling mistletoe on deciduous trees [120].

Chemical control of mistletoe poses some difficulties: lack of application technology, chemical damage to the host, reinfestation due to parasite seed germination, marginal crop selectivity, environmental pollution, low persistence and availability [50,119]. For these reasons there is a need for alternative control methods, such as biological control (biocontrol) [75].

Host compatibility is an important regulator of mistletoe abundance and distribution [121]. An alternative approach to protective management is the development of host cultivars resistant to mistletoe infection. Grazi et al. [122] found that some species (e.g., oak, larch, and elm) are rarely infected by *V. album* and that host resistance is genetically determined. Some cultivars of *Populus* are known to be resistant [123], because they possess natural physical barriers and resistance-conferring chemical compounds [124]. These include the number of fiber clusters and the thickness of suberized phellem cells produced towards the surface of branches by the phellogen that contain higher polyphenolic contents, which are induced in response to parasitic attack. The most resistant cultivar possessed the greatest number of polyphenolic cells per unit surface area. Later research with three *Quercus* species confirmed the same pre-existing anatomical and chemical features of resistance as those of *Populus*: thicker cortex, collenchyma, and fiber layer and greater density of polyphenol-containing cells [125].

Proper forest management and maintenance of a higher stand density are of great importance for controlling mistletoe infection. Mistletoe are favored on dominant trees due to higher light levels in the top of the crown and their generally larger root systems, that provide better access to water and mineral nutrients [109,121]. Also, larger trees usually have thicker branches, which provide a platform for birds, who may then deposit mistletoe seeds in their excrement, and also as a site on which mistletoe seeds can germinate. For these reasons, forest fragmentation and thinning favor the growth of *V. album* by increasing light levels [18].

## 5. Perspectives

"Prediction is very difficult, especially if it's about the future" (Niels Bohr). However, current projections indicate that climate change will increase weather anomalies, affect tree ranges, lead to increased threats from insects and diseases, as well as intensified droughts and hurricanes. It is also anticipated that climate change will increase the occurrence of mistletoe in pine stands. The spread of this parasitic plant will deepen the water stress experienced by trees, reducing their annual growth and accelerating crown dieback.

To appropriately manage the damage caused by mistletoe, it is necessary to have knowledge of the biology of this species, distribution mechanism, and relationships and interactions with host species. Evaluation of the impact of mistletoe on the annual growth of trees—and hence on the productivity of stands—requires monitoring. It should be based not only on the assessment of outbreaks of mistletoe (both from ground level and from the air), but also on the analysis of the impact on the health condition of individual trees, in relation to their age and site. This information can inform decision-making on when or whether to remove mistletoe or to remove trees infected by mistletoe.

It may also be asked whether mistletoe should be left to take its course without human intervention, since it is a natural forest element that could play a role in the future development of ecosystems as they respond to increasing stresses caused by climate change.

## 6. Conclusions

- We have proved that the damage caused by mistletoe have become an increasingly serious problem in pine stands.
- Mistletoe in Scots pine stands occurs mostly in areas with the highest percentage increase in spring air temperature and decrease in precipitation.
- Mistletoe infestation of pines leads to a decrease in growth and a deterioration in the health condition of trees.
- There are currently no effective and cheap methods of controlling mistletoe on an economic scale.
- Climate change can increase the occurrence of mistletoe in pine stands.

**Author Contributions:** All authors substantially conceived the idea, contributed in conceptualization, resources, writing the original draft, review and editing the text.

**Funding:** This paper was partially supported by the State Forest Holding in Poland (Project No. 500 426; contract number OR.271.3.4.2015).

**Acknowledgments:** The authors thank the Forest State Holding (DGLP) for field data and the anonymous reviewers for their valuable comments and suggestions.

**Conflicts of Interest:** Authors declare no personal circumstances or interests that may be perceived as inappropriately influencing the representation or interpretation of reported research results.

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
