# Peer review of "Impact of Common Mistletoe (Viscum album L.) on Scots Pine Forests—A Call for Action"

_forests, doi:10.3390/f10100847_

Round 1

Reviewer 1 Report

Impact of common mistletoe (Viscum album L.) on  European forests

Review:

General Comments:

I enjoyed reading this review of Viscum in European forests. Reviews like this are important to publish so they can inform the reader about the topic and increase cooperative research and appropriate action if needed. I would love a sub title of- “:a call for action” since you have very little data on impact and the paper is a review of the biology and management and points out needs.

 I think the review can be published with some major modifications to statements especially in abstract and section one as suggested below.

Most sections reviewed the literature well but reorganizational changes are needed in a couple places. 

The reader would have an easier time understanding the information  if the authors clearly state in the first sentence of each paragraph what the overall major conclusion or point the reader is to get from that paragraph. The authors do this for many paragraphs but not all.

Specific comments:

Line number:  Comment:

13     I find it hard to believe that Viscum has killed trees over thousands of ha.  Do you mean that it was found in the stands and maybe contributed during a severe drought?   Please make sure you do not say things that are not supported by direct data- such as data collected on many  trees showing for example that the dead trees had 100% of their branches infected and surviving trees had 0-10% of their branches infected. 

14      “Drought resulting from global climate change is implicated as an important factor conducive to the spread of mistletoe”  I think you must mean that the damage is increasing because of climate change.  How can the spread of the mistletoe be directly involved with climate-  unless birds numbers are increasing and migration is increasing because of climate change.  I understand that mistletoe can be creeping northward or something with changing climate but this parasite is not like a wind blown fungus that can spread across a continent in one season. Please review what you really want to say in the abstract since it currently makes no sense with the biology of this parasite.

37     I do not see any reason to list how many references there are on mistletoe since you are not analyzing the data from those abstracts or results of the research.  I would drop this paragraph.

42   Please avoid using the word  ”this”.  The reader always has to figure out what you were thinking.  So just say “within the “Viscum “species-----

72 Drop “However”  In Poland --- rephrase “  on roadside and park trees is common”

74  “ Short-term forecast of the occurrence of major pests and infectious diseases of forest trees in 2019" , between 2017 and 2018 the area of coniferous forest affected by mistletoe increased from 1.4 thousand to almost 23.0 thousand ha”.  No true mistletoe can spread this fast-  maybe you are thinking some other stress coupled with the mistletoe has caused visible damage to trees?   – If this increase is in the percent of infected trees the data must be an artifact of the survey system or human error or some other non-biological reason.   Wishy washy use of terms for serious forest damage is very confusing.   

 You would be wise to make sure you explain what “area affected means.”  Is this one tree infected per ha or what?  These are serious allegations of increase of area infested that do not make sense.  Please back up and figure out what the data are really about.   I also think you need talk about how to separate the percent of infected trees from the amount of the mistletoe on the trees from the resulting damage or the severity of the damage.

There are three data components to disease damage to trees:  1.  there is the number of trees infected= Incidence.  2. Then how much of the tree is infected= severity.   3. then there is the actual impact to the tree or amount of dieback or growth loss from none to 100% or death.  If you actually have these data items they need to be carefully explained  and then you can relate your regional climate and such to what is happening in the forests.

In a very general sense we use how many ha have an infected tree thus ha infested to give an indication where the pathogen is located.  This the first or coarsest data indication of a disease and can be useful to study the spread of a disease or insect but only when you link Incidence, severity and Impact can one get really excited about disease epidemiology data.

87  I do not think you mean that colonization is by pollen or by birds for that matter.  Birds are vectors or carriers of seeds.  I think what you are trying to say is that the weather conditions are probably conducive for pollination, seed dispersal by birds or ?

92  Please define what “area of Scots pine stands affected by mistletoe” means.  Is this one tree infected per ha?  Is this total death of all trees killed somehow by Viscum? I would doubt that.  The uniformed reader will jump to all sorts of conclusions without some better definition of your terms.

97  I think this section is really better titled something like Viscum Biology and  Host Impact

I would reorder this section so that the reproduction and general biology of the parasite and how it spreads is first and then the impact on the host follow.

121  I think you need an expanded paragraph here explaining how the seeds spread from tree to tree and within a tree. Birds move seeds long distance and sticky seeds are short distance?  I assume this is the story?

This paragraph should not contain information on endophytic system-  that information should be another paragraph with topic sentence- colonization of host tissue- or something like that

148  This is because the birds land at top of tree and infections start there and then sticky seeds may drop or be moved downward?  This sentence should be part of a paragraph under section two:  Spread of Viscum is accomplished by --   You can move the first paragraph from section 3 to this paragraph so all the biology of the parasite is in one section.

151-162   This section on the spread biology of the mistletoe is written well and adds to the paper and allows reader to learn about the issue but should be moved back to section two and incorporate the information on sticky seeds and how they move.  Do they move at all on their own – i.e. fall down or are they totally moved by birds???

163-205   This section three can now just become about potential bio control agents.

186-2005  These paragraphs are about the impact of Viscum or mistletoes on the plant so they belong in Section 2 under impact on the host. They do not belong here.

295-309  This section very important to make sure you are suggesting very important research steps that should be taken.  I think you have a done a good job with this paragraph.   You can suggest that all aspects of Viscum ecology ie what positive aspects are there of infestations- ie bird food- needs to be considered so management actions are appropriate. 

Author Response

Added as an attachment

Reviewer 2 Report

I think that readers in Forest will be very interested in the topic. The manuscript explained the impact of milestones on forests based on a number of references as a review paper. I believe, the manuscript is composed and written well. I recommend only one to the authors. If there is a table or a picture that summarizes what you explained, it will help the reader better understand.

Author Response

Added as an attachment

Reviewer 3 Report

This manuscript is a review analysing the impact of a semiparasitic plant (the mistletoe Viscum album) in European forests. The topic is of interest and the revision is pertinent. Yet, it would require some work before eventual publication. The authors need to make an effort to better frame and conduct ideas along the manuscript, which can be done with a revision of the text. The text and flux of ideas needs to be better organized along the manuscript.

I really miss a much deeper discussion on the effect of climate change (and also global change) on mistletoe dynamics. This is only mentioned in the last subsection (5), but merits further discussion, probably with a subsection exclusively dedicated to the topic. I would also recommend including (and probably start) one of more subsection/s more clearly describing the ecology and etiology of the semi-parasitic species (now this information is on subsections 2 and 3). In general, titles of the different subsections should be more informative and precise (e.g. line 151). Mistletoes are not new and play a role in forest ecoystems. The problem is to know whether infestation is increasing owing to climate or global change (as they discuss in the article, e.g. lines 101-103) and if this threats forest activities. This point should be better highlighted in the abstract and the manuscript. For instance in the abstract it is mentioned that it affects “deciduous” species, and then the manuscript refers mostly to conifers. The abstract can be more informative in general.

As a minor comment, to help readers better understand and frame the manuscript the authors also should be more descriptive when they only refer to Polish datasets (e.g. Figure 2) and rather than referring only to the country where data belong to, they could add some insight on the ecology of the discussed data: e.g. temperate, Mediterranean, boreal, xeric, mesic, etc. Point 4 is fine … but it must be discussed in relation to climate change issues: control is secondary and useless if there is a dominant environmental factor behind mistletoe infection and mistletoe is just a secondary agent helping to produce species decline.

- Line 11, line 73: why need to constrain only to managed stands? Mistletoes are also present in unmanaged stands, or do authors suggest that higher mistletoe infection rates are a consequence of management?

- Line 12: losses of what?

- Line 16: semiparasitic, it is able to photosynthesize (as mentioned later in the manuscript).

- Line 63-64: is it because mistletoes respond to warming or because as a consequence of warming the availability of hosts is increasing and hosts are ascending in altitude? This is important and needs to be discussed in the ‘ecological’ subsections.

- Line 66: what do authors mean about ‘biotic losses’?

- Line 85: please, define ‘moderate’; do authors refer to humid stands? Average in terms of water availability within the host distribution? It is interesting to be coherent with Figure 2, where the authors nicely show a relationship between mistletoe infection and drier/warmer conditions.

- Line 87: mistletoe colonization by ‘pollen and seeds’? This does not make sense. Mistletoe colonizes mostly with the aid of birds, as the authors discuss (line 153).

- Line 89: please specify ‘in Poland’ (I guess that map is from that country).

- Line 92-93 and Figure 2: very nice result; please explain briefly the data you use to make these inferences.

- Line 106: there are more references and more effects of mistletoes in the physiology of hosts that need to be acknowledged here. See e.g. Zweifel et al. (2012) (ref 94).

- Line 107: add ref 94 too.

- Line 149: do authors refer here to tree age or mistletoe age?

- Line 154: B. garrulus, please correct.

- Line 163: what do authors mean? That they feed on mistletoe, that they are a biological control for mistletoe?

- Lines 175-180: are those species only found in Poland? Please, try to frame within Europe your results (as expressed in the title).

- Line 186-188: please, explain better.

- Line 190: richness of what?

- Line 274-276: then, all the insects and fungi referred before in subsection 3?

- Line 294: even if this is true probably, thinning are regularly applied (and necessary) in silvicultural schemes, and are also receiving much attention in terms of mitigation of water stress in the face of climate change. Please, as I suggest above, better discuss ideas in relation with environmental factors’.

- Line 297: delete ‘s’.

Author Response

Added as an attachment

Round 2

Reviewer 3 Report

The authors addressed properly all comments from the previous revision. They have now constrained the title of the study only to Pinus sylvestris, I guess this was suggested by other reviewer.

Just some very small considerations from my side:

- Line 36: add ‘defoliated’ to weakened, or substitute it.

- Line 60: please add, ‘likely’ to ‘it likely happens’. Not only the direct effect of temperature, but likely also the fact that with climate change there will be more susceptible hosts at warmer sites can explain the shift in altitude.

- Line 92: specify the period when you calculate the mean.

- Line 102: please, explain what are the data depicted in that figure, plots as from Figure 2?

- Line 159: what is ‘viscumriose’? New term for what and why?

- Line 167-168: no need to speculate and introduce the concept ‘hydraulic failure’ there, that is not implicitly addressed in the reference provided.

- Line 181: specify that ‘dodder’ refers to Cuscuta sp.

Author Response

Reply to the review in the attached file
